



# Extraction of spatially confined small-scale waves from high-resolution all-sky airglow images based on machine learning

Sabine Wüst[1], Jakob Strutz[2], Patrick Hannawald[1], Jonas Steffen[3], Rainer Lienhart[4], Michael Bittner[1, 5]

[1] Erdbeobachtungszentrum, Deutsches Zentrum für Luft- und Raumfahrt Oberpfaffenhofen, 82234 Wessling, Germany

[2] formerly at Erdbeobachtungszentrum, Deutsches Zentrum für Luft- und Raumfahrt Oberpfaffenhofen, 82234 Wessling, Germany, now at Infineon Technologies AG

[3] formerly at Institut für Physik, Universität Augsburg, 86159 Augsburg, Germany

[4] Institut für Informatik, Universität Augsburg, 86159 Augsburg, Germany

[5] Institut für Physik, Universität Augsburg, 86159 Augsburg, Germany

*Correspondence to*: sabine.wuest@dlr.de

## Abstract

Since June 2019, a scanning airglow camera is operated operationally every night at DLR Oberpfaffenhofen (48.09°N,

11.28°E), Germany. It provides nearly all-sky images (diameter 500 km) of the OH* airglow layer (height ca. 85–87 km) with an average spatial resolution of ca. 150 m and a temporal resolution of ca. 2 min.

We analyse about three years (941 nights between October 2020 and September 2023) of OH* airglow all-sky images for spatially confined wave structures with horizontal wavelengths of ca. 20 km and less. Such structures are often referred to as ripples and are considered to be instability structures. However, Li et al. (2017) showed that they could also be secondary

waves. While ripples move with the background wind, secondary waves do not.

To identify small-scale and spatially confined structures, we adapt and train YOLOv7 (You Only Look Once, version 7), a machine learning approach, to determine their position and extent on the sky as well as their horizontal wavelength. Those wavelengths are compared to two-dimensional FFT (Fast Fourier Transform) results. We analyse the seasonal variations in the propagation direction and horizontal wavelengths of these structures and deduce that instability signatures are observed

especially in summer.

Finally, we introduce a concept for "operating-on-demand" in order to derive energy dissipation rates from our measurements.



## 1 Introduction

About 100 years ago, the radiation of the hydroxyl (OH*) airglow layer was observed from the ground for the first time without knowing which molecule caused this radiation until Meinel (1950) identified OH* as the source. While he exposed photo plates for approx. 4 h, today two-dimensional InGaAs (Indium-Gallium-Arsenid) sensors manage with exposure time of the order of a second to get a sufficient OH* signal. Due to technical progress, OH* airglow observations, which refer to ca. 85–87 km height with some seasonal and geographical variation (Wüst et al., 2017, 2020; Savigny, 2015; Baker and Stair, 1988), have now been used for several decades to analyse gravity waves (GWs, see, e.g., López-González et al., 2020; Wüst et al., 2018; Wüst et al., 2016; Offermann et al., 2009; Taylor, 1997; Takahashi et al., 1985) whose intrinsic frequencies are lower than the (angular) Brunt-Väisälä frequency, which is ca. $2.0$–$2.3 \cdot 10^{-2}$ s$^{-1}$ (= 4.6–5.2 min) for the OH* airglow layer with some seasonal and geographical variation (Wüst et al., 2020).

GWs are of great interest because they contribute significantly to the vertical and horizontal redistribution of energy and momentum in the atmosphere. However, it is still difficult to measure where and how much energy they deposit: the spatial resolution of today's instruments for observing turbulence is often insufficient. In 2021, Sedlak et al. (2021) and Hecht et al. (2021) were able to observe the breaking of GWs using two-dimensional OH* airglow measurements. The spatial resolution of their observations is in the range of ca. 24–25 m / pixel, the derived energy dissipation rates are 0.08 and 9.03 W/kg. In order to achieve such high spatial resolutions, it is necessary to focus on a relatively small area of the sky (13.1–14.1 km × 13.4 km in Sedlak et al. (2021), 47.4 km × 47.4 km in Hecht et al. (2021)) near the zenith. At the same time, the observation of an extended area (preferably all-sky) would be desirable to derive as much information as possible.

One, albeit not perfect, way to achieve this is to operate a scanning camera that measures all-sky but at the same time has the temporal and spatial resolution to at least recognise small-scale wave-like signatures, which are referred to as ripples in literature. Ripples are characterized by horizontal wavelengths of 5–15 km (Li et al., 2005; Taylor et al., 1995) or of 20 km at most (Takahashi et al., 1985). Their lifetime is in the range of 45 min or less (Hecht, 2004). In the past, such small-scale structures were often attributed to instability signatures, which may be related to or also be part of a breaking process of a GW (e.g. Li et al., 2005; Hecht, 2004; Fritts et al., 1997). If this is the case, they are supposed to move with the background wind (Hecht, 2004; Fritts et al., 1997). Their orientation with respect to the generating GW is sometimes interpreted with respect to the generating process (convective or dynamical instability). Today, it is known that such small-scale structures, especially if not moving with the prevailing wind, can also be secondary waves (Li et al., 2017). Unfortunately, wind information is often not available at OH* airglow measurement sites. This makes it difficult to differentiate between secondary waves and instabilities. However, in order to distinguish between both features, we propose to align a second camera with a higher spatial resolution to the ripple. If turbulence is observed, this is not only an indication of a breaking GW, the size and speed of the vortices relative to the surroundings can also be used to provide information about the dissipated energy (Sedlak et al., 2021). The first step in implementing this concept is the operation of such a scanning camera and the rapid analysis of its data with regard to small-scale wave-like structures in the spatial range of ripples. This part is the focus of this manuscript. Since the



term ripples is associated with instability structures in literature, we call the analysed signals small-scale wave-like structures in the following.

At DLR Oberpfaffenhofen (48.09°N, 11.28°E), Germany, a scanning camera has been operated since 2019. It is based on the FAIM (Fast Airglow IMager) instrument already presented in Hannawald et al. (2016). Operated in the scanning mode, as presented here for the first time, it takes 35 pictures in consecutive order of different azimuth and zenith positions, which together cover an area of ca. 500 km diameter over Oberpfaffenhofen with an average spatial resolution of approx. 150 m/px. One complete scan takes 2:03 min. Every night, it delivers approx. 1.5–2.0 GB of image data. Due to this large amount of data,

machine learning algorithms have already been tested or used to distinguish cloudless from cloudy or dynamically quiet from dynamically active episodes in camera images (Sedlak et al., 2023). However, in all these cases the cameras were static.

This publication now presents a machine learning method to identify spatially confined wave-like signatures in the scan images and to derive their horizontal wavelengths. Before machine learning became popular, such analysis was mostly done by eye, i.e. difference images were calculated to remove stationary or nearly-stationary structures or to reduce their signal. The wave

signatures were then identified and characterized by eye (Li et al., 2017; Suzuki et al., 2011; Yue et al., 2010). We rely here on a modified YOLOv7 (You Only Look Once, version 7, Wang et al. (2022)) algorithm to identify such signatures and characterize them with respect to horizontal wavelength and propagation direction. Already Lai et al. (2019) presented a machine learning method to detect spatially confined wave signatures in OH* airglow images derived by a static camera. Those authors use a two-stage method, i.e. a classification model based on convolutional neural network (CNN) for the

selection of clear-sky images and an object detection model based on faster region-based convolutional neural network (Faster R-CNN, Ren et al. (2017)) for the localization of wave patterns. The horizontal wavelengths were then derived using a two-dimensional (2D) FFT. In contrast to their work, YOLO is a one-stage detector, i.e. the objects are identified and classified in one step, which makes it faster than two-stage detectors; at least that used to be the case. The development in this thematical area is very fast and today's versions of the algorithms have improved to such an extent that there is less difference in time.

We compare the accuracy of our modified YOLO algorithm with the accuracy of a 2D FFT. Furthermore, we use information about the propagation direction and the orientation of the wave-like structure to discriminate to some extent between secondary waves and instability features.

This paper is organised as follows. Section 2 introduces the scanning FAIM instrument and the pre-processing of the raw data. Section 3 contains a detailed description of the YOLOv7 algorithm and its modifications. Section 4 presents the results and

section 5 their discussion. The paper ends with a summary and outlook in section 6.



## 2 Data basis

### 2.1 OH* airglow imager

The FAIM 4 camera is a 320 pixel x 256 pixel InGaAs-camera. The air-cooled sensor operates at 235 K. The full field-of-view of an individual image is 22.5° (vertical) x 18.1° (horizontal) with the lens having a focal length of 23 mm. The integration time of the camera is 0.5 s. It is mounted on one vertically oriented rotation stage (zenith angle stage) and one horizontally oriented rotation stage (azimuth angle stage), which allow to point the camera to any position of the sky (see Figure 1). The default operation is to scan the sky to provide a high spatial resolution composite image from multiple individual images, with

high spatial coverage.

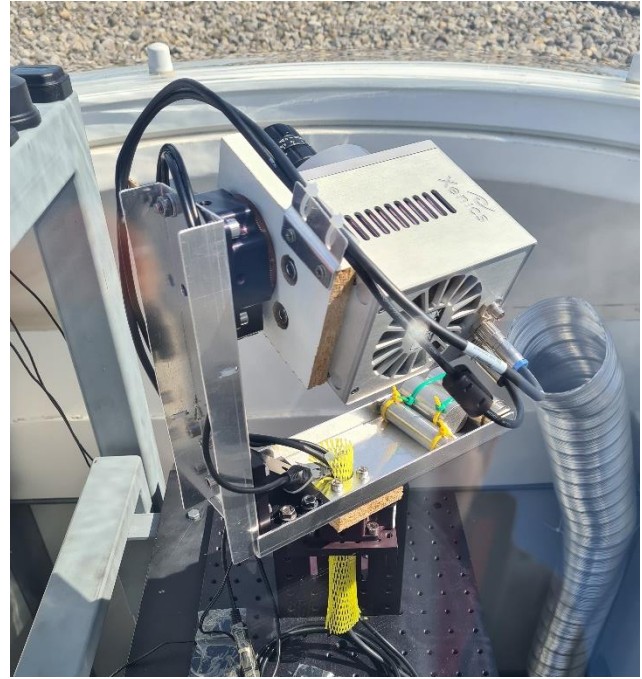

**Figure 1 Photo of the scanning FAIM instrument. The azimuth rotation stage is below the horizontal metal frame, the second stage is vertically oriented to provide the possibility to rotate the camera to different zenith angles. The construction is counterweighted to prevent for unilateral strain and provide a longer lifetime of the mechanical components. The instrument is located below an air-**

**ventilated acrylic dome at Oberpfaffenhofen.**

Due to the time required to rotate the instrument between images, it takes 2 min 3 s to complete a scan consisting of two full circles with zenith angles of 60.3° and 38.3°. These values are set so that there is minimal overlap between the images. The images of a completed scan address zenith angles from 27.5° to 71.5°. Due to the high precision and repeatability of the stage positions, there is no need to have an overlap between the images to stitch them together, as is often done in photography.

Each of the 35 individual images is unwarped into a trapezium shaped equidistant image as described in Hannawald et al. (2016), and stitched together into a 2600 x 2600 scan image based on their azimuth and zenith positions. The pixel grid for this unwarping is set to a resolution of 200 m/px. There is no further processing regarding the stitching, which results in a





small seam between the images, but this is handled correctly by our machine learning algorithm. As a result of using only two circles, there is a hole with no information in the centre of the scan image from the zenith angel of 0° to the zenith angle of

27.5°.

Figure 2 shows the spatial resolution per zenith angle of the scanning FAIM (before unwarping) compared to typical OH all-sky imagers. The resolution of the scanning FAIM ranges from 140 m/px at 30° zenith angle to about 900 m/px at 70° zenith angle. The chosen resolution for the unwarped pixel grid of 200 m/px therefore results in averaged image information for the zenith angles of about 43° and smaller, as multiple pixels of the original images are remapped to one pixel in the unwarped

scan image. For higher zenith angles, the image information is interpolated as usual in this type of process.

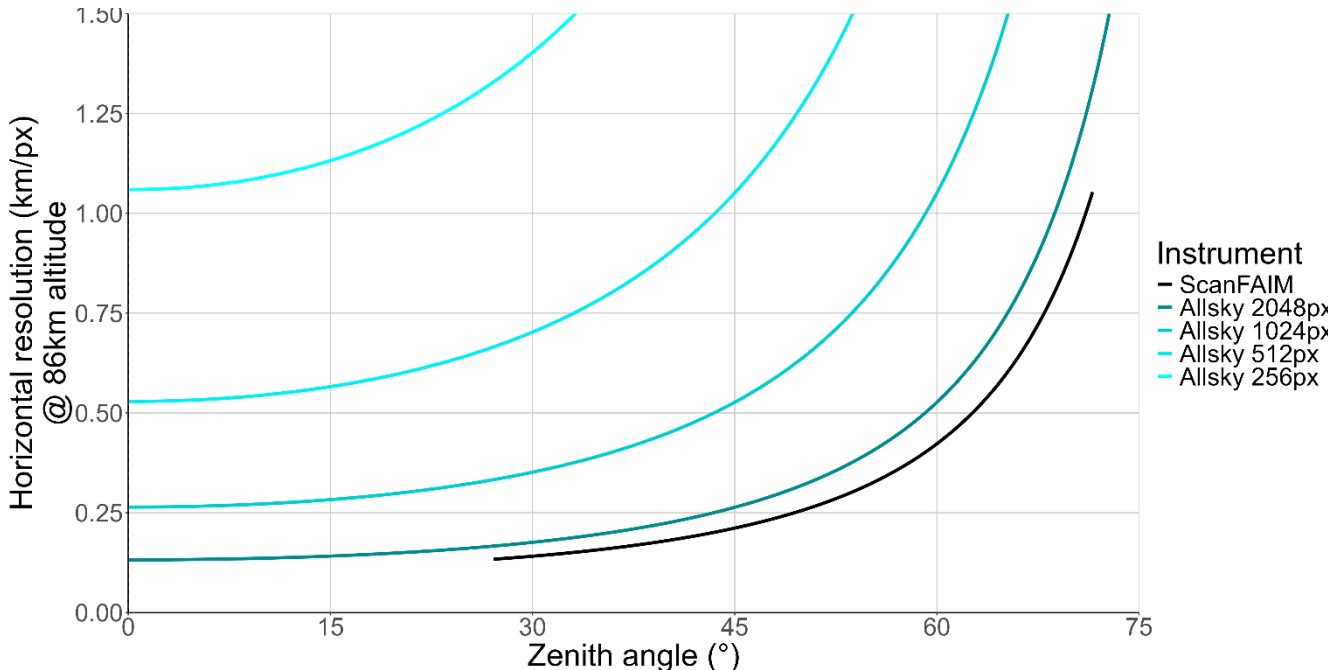

**Figure 2 Horizontal resolution per zenith angle for typical OH all-sky imager systems (turquoise) and the scanning FAIM system (black). Many SWIR all-sky imagers have image circles, i.e. illuminated area of the sensor, of about 500 pixels, while NIR all-sky**
**imagers often have 1,000 pixels or in rare cases also 2,000 pixels image circles.**

A third (inner) circle would result in a high level of redundancy due to highly overlapping parts, while taking a much longer time to complete the scan image (approx. 3 min). Furthermore, the waves in the centre of the inner circle could move quite a lot during the time it takes to scan the inner circle, which would cause stitching artefacts. This does not happen – or only to a very small extent – when using only the two outer circles: assuming a fairly fast gravity wave with an observed phase velocity

of 100 m/s and an average time of 3.5 s to acquire the adjacent image with a resolution of about 150 m/px, there could be a mismatch of up to 3 pixels, normally it is much less. Between the inner and outer circles, it could be more, as one circle is



completed before the second is started (in reverse order). Figure 3 shows an example of a scan image with high wave activity. The corresponding video sequence of the night is in the supplemental data of this manuscript.



**Figure 3 Sample of a OH nightglow scan image composed of 35 individual images. The circle has a diameter of about 520 km (at 86 km altitude) and covers zenith angles from 27.5° to 71.5°. The average resolution is 340 m/px. Several GWs with horizontal wavelengths of the order of 20 – 30 km are visible as well as many structures with horizontal wavelengths smaller than 10 km.**




## 2.2 ERA-5

Since clouds are an obstacle for ground-based SWIR measurements, cloud cover data above FAIM 4 are extracted from ERA5 (European Centre for Medium-Range Weather Forecasts Reanalysis, fifth generation) and analysed to exclude cloudy episodes from further analysis. As delineated by Hersbach et al. (2020), this reanalysis data set features hourly data and a horizontal resolution of $0.25° \times 0.25°$. At Oberpfaffenhofen, this is approximately 27.5 km in meridional direction by 18.5 km in zonal direction. The nearest data point to FAIM 4 is at 48°N and 11.25°E. This is the only one used here, as the observed area at low atmospheric heights is covered by a single grid cell. The cloud cover is determined by identifying the maximum value among low, medium, and high cloud cover layers. This methodology is chosen since atmospheric water vapor, regardless of its altitude, poses a significant impediment to the camera's observational capabilities. In comparison with the aggregated cloud cover data from ERA5, this approach shows better agreement with the cloud cover observed by eye in the FAIM data. This is based on the comparison of several nights.

The ERA5 data set is generated using an assimilation model, and cloud cover is just one of many globally consistent data products that are the output of this assimilation model (see Hersbach et al. (2023) for further information on ERA5 and its availability). Wu et al. (2023) compare the total cloud cover and the high cloud cover between the satellite MODIS (Moderate-resolution Imaging Spectroradiometer) and the four reanalyses ERA5, ERA-interim (ECMWF Interim Reanalysis), MERRA-2 (Modern-Era Retrospective Analysis for Research and Applications, version 2), and NCEP (National Centers for Environmental Prediction, here only the total cloud cover). They conclude that ERA5 has the highest correlation with MODIS among the mentioned reanalysis data sets for total cloud cover for latitudes between ±60°, based on monthly means. The root mean squared errors for the different months vary between 0.080 and 0.098. All these mentioned models perform better over land than over sea.



## 3    Analysis methods

In order to achieve the results shown in section 4, a neural network is trained. Due to its speed and accuracy ((Meng et al., 2023), (Zhao et al., 2023)), YOLOv7 is chosen for this purpose.

A neural network in general consists of different nodes arranged in layers. Each node receives different inputs, e.g. from other nodes in upstream layers. These inputs are weighted and added. A predefined non-linear activation function is applied to the result. The output of this activation function is forwarded to other nodes in subsequent layers, where the procedure is the same as just described. As part of the training of a neural network, the weights of the various nodes are optimised against a loss function until the output of the neural network is sufficiently good. This is therefore a non-linear optimisation problem.

The weights can be chosen arbitrarily at the start of training or a pre-trained network can be used. The advantage expected from a pre-trained network is that the optimisation problem can be solved more quickly. In addition, better results have also been observed in some cases (Hendrycks et al., 2019).

BYOL (Bootstrap Your Own Latent), an unsupervised learning approach, is used here for pre-training, i.e. to determine an initial set of weights for the YOLOv7 backbone, which consists of all but the last layers of the YOLO network. In these

backbone layers, specific features of pictures are identified. In the last layers of such a network, in the so-called head, such features are used to identify the object, which is present in the picture. In the work by Sedlak et al. (2023), a neural network was trained for FAIM 3 (static camera in Oberpfaffenhofen) that can distinguish between dynamically quiet, dynamically active and cloudy episodes. The field of view of FAIM 3 lies within the field of view of FAIM 4. For this study, BYOL is trained with 10,000 FAIM 4 images that were dynamically active according to the FAIM 3 analyses of Sedlak et al. (2023).

In contrast to 'classical' object detection applications (e.g., looking for a cat in a picture), the small-scale wave-like structures we focus on here are less clearly separated from the background (Figure 4). Therefore, it is often difficult to precisely define their boundaries. This makes it equally difficult to use conventional algorithms to detect the direction of propagation.

In the following subsections, the basic idea of YOLOv7 is described as well as its modifications for the identification of horizontal wavelengths and orientation. Different quality measures for YOLOv7 are presented and the derived wavelengths

and the orientation of the wave fronts are compared to results based on a 2D-FFT. Finally, the algorithm applied for the derivation of the propagation direction of the identified small-scale structures is introduced.





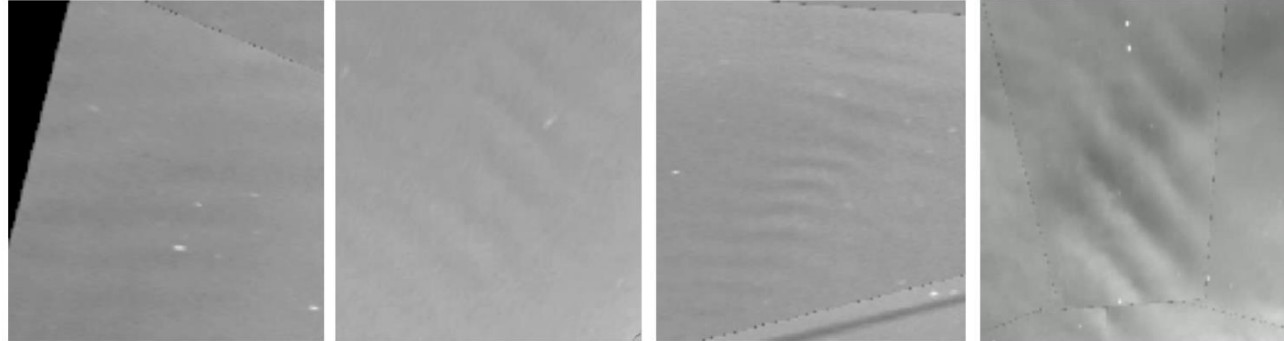

**Figure 4 Wave-like structures in airglow imager data are characterised with varying degrees of clarity. The structures in the two images on the left are very faint, while the images on the right show clearer indications of such structures. This classification is subjective. The solid line in the bottom right-hand of the third image and the dashed lines on all images are artefacts. The dashed lines are the edges where the images are stitched together, the solid line is a lightning rod.**




## 3.1    Description of YOLOv7 and its modifications

YOLOv7, an evolution of the YOLO (You Only Look Once) object detection architecture, operates by gridding the input images (so defining knots and edges) and directly predicting bounding boxes (where a specific object is present) and class probabilities (which object is present) for each knot of the grid. It utilizes a backbone convolutional neural network (CNN) for
feature extraction and incorporates a feature pyramid network (FPN) to capture multi-scale features: the features can be interpreted as image patterns, relevant for detecting the wave-like structures, while the multi-scale features are essential for detecting objects of varying sizes. The detection head of YOLOv7 predicts multiple bounding boxes with associated confidence scores for each knot of the grid, followed by non-maximum suppression (Neubeck and van Gool, 2006) to refine the final detections. This is required, because there might and most likely will be multiple predicted bounding boxes for the
same object. Non-maximum suppression removes all boxes for the object, except the one with the highest confidence score. This architecture enables real-time object detection in different applications.

According to Wang et al. (2022), the authors of YOLOv7, this algorithm exceeds the performance of all object detectors known until then  in terms of both speed and accuracy. They also claim that YOLOv7 achieves the highest accuracy among all known real-time object detectors operating at 30 FPS (frames per second) or higher.

As already mentioned above, we not only use YOLO in order to detect wave-like structures, we also modify the algorithm to derive the horizontal wavelength and the orientation of the wave-like structure. The motivation for developing modifications to the YOLOv7 algorithm is addressing the limitations associated with the 2D Fast Fourier Transformation (2D-FFT), which is often used in scientific papers for analysing wave patterns in (airglow) images. Although powerful in certain contexts, the 2D-FFT faces two main challenges. Firstly, performing a 2D-FFT, especially on high-resolution images, is time-consuming
and computationally expensive, leading to longer processing times and significantly affecting efficiency in analysing large data sets. The need for quick and accurate results, especially in real-time applications or when processing large data sets, requires the exploration of alternative approaches such as artificial neural networks, which are time-consuming to train but not to use. Secondly, the 2D-FFT is susceptible to strong contrast variations within images. The specific scan geometry of FAIM 4 captures results in high contrasts at the edges between the individual pictures and the black background, which can lead to
wrong wave-parameter calculations of the 2D-FFT. Moreover, the 2D-FFT cannot distinguish between image information caused by airglow and artificially induced information such as black image borders, lightning rods within the field-of-view, and other artefacts. In the case of the scan images shown here, the black circle in the zenith orientation, for example, prevents useful 2D-FFT calculations when the analysis is applied to the entire images. The solution to this problem is to calculate the 2D-FFT on a number of sub-images that exclude any artificial signals. However, this approach results in the loss of some
information and requires the results of the potentially overlapping sub-images to be algorithmically combined to ensure the integrity and accuracy of the final wave parameter calculations. In addition, the edges within a FAIM 4 image, due to the successive scanning of the sky, would also cause additional artificial signatures. The modifications to YOLOv7 aim to address



these issues by providing a more efficient and robust method for detecting and analysing wave signatures in images. By directly integrating wavelength and orientation prediction into the object detection process, the modified algorithm bypasses the need

for a 2D-FFT analysis.

A crucial part of working with machine learning is training the algorithm, so determining the weights of the various nodes of the neural network. For this, labelled data are needed. This labelling is done manually. The labelled data set is then separated into 80% training data and 20% testing data. We use 160 images without any wave-like structure and identify more than 1,850

wave-like structures on 1,000 additional images. In principle, images without marked objects are not necessary, but some are inserted to train the neural network from certain errors. For example, the sunrise leads to some kind of periodic artefacts in the image, which should not be interpreted as atmospheric structures. As there are no images showing atmospheric wave-like structures during sunrise, this possible error is counteracted by inserting sunrise images without marked boxes into the training data set. In order to modify YOLOv7, the labelling process is extended to include information about the wavelength (measured

in pixels) and an angle of wave orientation (ranging from 0° to 180°). Overcoming the challenge of cyclicity of the wave orientation prediction (in particular the issue of proximity between angles near 0° and 179°), requires the transformation of the angles into sine and cosine. This ensures a consistent and continuous representation of this information. The transformations employed are as follows: forward conversion with $sin\left(\alpha \cdot \frac{\pi}{180}\right), cos\left(\alpha \cdot \frac{\pi}{180}\right)$ and reverse conversion utilizing $atan2(y, x) \cdot \frac{180}{\pi}$. The atan2(y, x) is defined as follows:


$$
atan2(y, x) = \begin{cases} \arctan\left(\frac{y}{x}\right) & \text{if } x > 0, \\ \arctan\left(\frac{y}{x}\right) + \pi & \text{if } x < 0 \land y \geq 0, \\ -\pi & \text{if } x < 0 \land y < 0, \\ +\frac{\pi}{2} & \text{if } x = 0 \land y > 0, \\ -\frac{\pi}{2} & \text{if } x = 0 \land y < 0, \\ \text{undefined} & \text{if } x = 0 \text{ and } y = 0. \end{cases}
\tag{1}
$$

To predict the wavelength with YOLOv7, the methodology is chosen analogously to the determination of box sizes (so in which part of the image a wave is present). YOLOv7 uses three different grids and predicts the box size relative to the grid size. The box width (which is equal to the box height) can be four times the grid cell width at maximum. The maximum

wavelength is limited to the size of the box itself, as a detected wave should have at least one crest and one trough within a box.

The main change of the algorithm is made to the output structure. For each detected object, the output is expanded to incorporate two additional values for the angle (sine and cosine) and one for the wavelength. As a result, the output dimension



per image is augmented from (100,800; 6) to (100,800; 9), seamlessly integrating the wavelength and orientation data without

significantly increasing the computational load. 100,800[1] is the number of predicted bounding boxes and the default YOLOv7 value for our image input size. These modifications result in a negligible increase in the total number of trainable parameters, with only a 0.2% increase in the overall model size. This maintains the computational efficiency of YOLOv7, demonstrating the feasibility of extending its application scope without compromising processing efficiency. Through these targeted modifications, we demonstrate an innovative adaptation of the YOLOv7 architecture, extending its utility beyond conventional

object detection to encompass the analysis of specific object characteristics such as wave parameters.

The training is based on a batch size of 16 images (i.e., 16 images are processed by the neural network together) and 1,000 epochs until precision and recall do not improve any further. To increase the number of images to learn from, the images are "augmented", i.e. a given image is transformed in a known way, e.g. rotated, scaled, etc. The standard augmentations of the

YOLOv7 algorithm are used, except for the so-called "mosaic" augmentation, as the wave structure is often no longer visible here. In addition, the BBoxSafeRandomCrop augmentation is applied. This cuts out a random area of the image, but ensures that no labelled objects are lost in the process. All augmentations that change the size of the image or rotate it must be rewritten in order to calculate the wavelength or wave orientation accordingly. This includes horizontal and vertical reflections of the image, rotations and the BBoxSafeRandomCrop augmentation.


In order to check the performance of YOLOv7, precision (ratio of correct positives to the sum of correct and false positives, this indicates how many detections were correct) and recall (ratio of correct positives to the sum of correct positives and false negatives, i.e. the ability of the model to identify all wave-like structures in the images) are calculated from the testing data. Depending on the objective, it must be decided whether a higher precision or a higher recall is important for the problem at

hand. In the case of this work, a good recall is relevant because statistics is to be made about the wave-like structures found and therefore a high number of wave-like structures is required. However, the results should not be falsified by misinterpreted cloud structures or similar events. A good precision is therefore also important. Precision and recall are calculated from the position of the wave-like structures, so from their bounding boxes. The predicted and the labelled bounding boxes are regarded as identical, if the IoU (Intersection over Union, i.e. the ratio between the overlapping regions of the predicted and the labelled

box to the region of the predicted and the labelled box together) value is greater than 0.5. As already explained above, the labelling of wave-like structures is somehow subjective. Therefore, precision and recall cannot be directly compared with values from other non-airglow studies and manual control of the results is necessary in any case. For this reason and due to the fact that the data set is not particularly large, we used the default hyperparameters of YOLOv7 and could therefore avoid

---

[1] The number of 100,800 predicted bounding boxes is calculated as follows. The image to be analysed is gridded into $40 \times 40$, $80 \times 80$, and $160 \times 160$ boxes, each box is assigned to one grid point. For each gridding, three FPN heads, which determine the size of the object, are calculated. So, we get $(40^2 + 80^2 + 160^2) * 3 = 100,800$ bounding boxes.



to use a validation dataset. The dataset was just split into training and testing data. The precision-recall curve is shown in

Figure 5. A confidence value of 0.5, which is the default value of the YOLOv7 implementation, leads to a precision and a recall of ca. 68% both. For some measurement nights, the confidence values are varied and the results are compared. A confidence value of 0.5 gives reasonable results in terms of the above-mentioned trade-off between recall and precision.

For the newly introduced parameters wavelength and orientation of the wave fronts, 78% are correctly identified. Correct in this case means that the angle of the orientation and the horizontal wavelength differ by less than 10° and 2 km (ca. 10 pixels)

respectively from that of the label. These tolerances have been determined empirically: the drawn wave crests still describe the wave relatively accurately despite these deviations. If one of the parameters deviates more than the given tolerances, both parameters are no longer considered correct. The ratio is calculated by dividing the number of correctly identified parameters by the number of identified parameters.

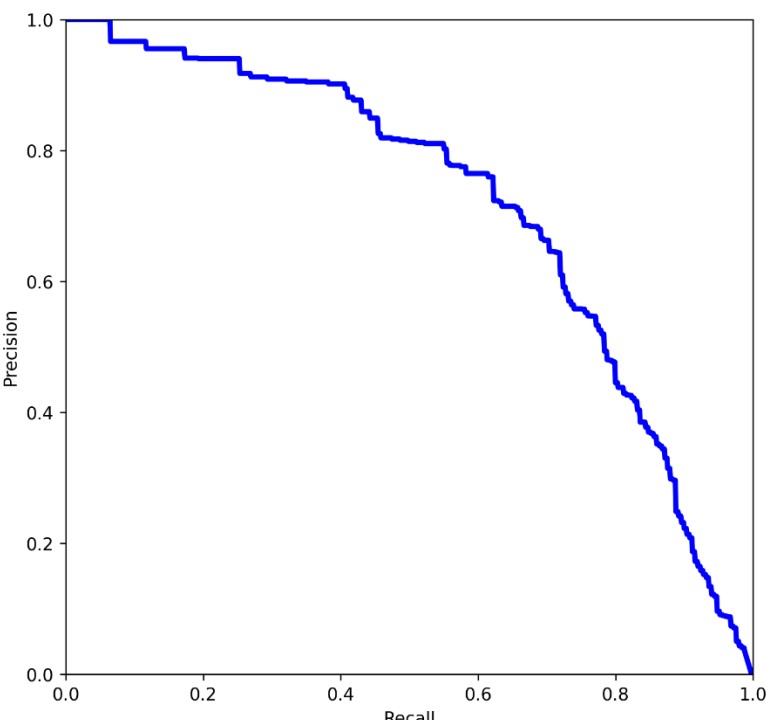


**Figure 5 Precision-Recall plot after the YOLOv7 training.**





### 3.2      Comparison of wave characteristics derived from YOLOv7 with 2D-FFT based results

The wave parameters of our modified YOLOv7 algorithm are evaluated against the results of a 2D-FFT derived from a tool previously used in (Hannawald et al., 2019) for wave analysis. In the following, the labelled data is always considered as the ground truth and any deviation as an error.

In a first step, the 2D-FFT is applied to the bounding boxes of the labelled data. Wavelengths greater than the maximum or less than the minimum over all the labelled wavelengths are discarded. Then, the remaining results' largest amplitude waves

are considered for comparison. For this purpose, the angle outputs from the FFT are adjusted modulo 180° to address the ambiguity between, for example, northward- and southward-oriented waves in individual images. The comparison (Figure 6) reveals that 78% of FFT predictions have a deviation less than or equal to 2.5° for the orientation and 3% (relative to the labelled wavelength) for the wavelength. 89% are within a 20° tolerance for the angle and a 21% (relative to the labelled wavelength) tolerance for the wavelength. This data suggests that the 2D-FFT tool generally provides relative accurate

predictions for detected wave parameters, if the bounding boxes are provided. However, overlaps between theses boxes with the black areas in FAIM 4 images lead to significant errors in 2D-FFT analysis: a strong increase in error for both wavelength and angle predictions can be observed in this case, emphasizing the need for pre-processing or the exclusion of such boxes from 2D-FFT analysis.

In a second step, the wave parameters derived by the modified YOLOv7 algorithm are compared to the labelled data. However,

this is not straightforward because 80% of the labelled data has already been used in training, and furthermore the predicted boxes do not exactly match the labelled ones. For the comparison, we therefore use the remaining 20% of the data. If we interpret bounding boxes with an Intersection over Union (IoU) value greater than 0.8 as sufficiently correctly identified, we achieve an accuracy of 78% for the wave parameters. The wave parameters are considered correct, if the wavelength error is less than 10% relative to the labelled wavelength and the error of the angle is less than 10°.

In summary, the 2D-FFT provides more accurate results as 78% of the FFT predictions have an error of less than or equal to 2.5° for the orientation and 3% (relative to the labelled wavelength) for the wavelength. For the modified YOLOv7 algorithm, 78% of the results are considered correct, if the wavelength error is less than 10% relative to the labelled wavelength and the error of the angle is less than 10°. However, this is only true, if the boxes do not overlap with the black areas of the image. In the case of overlaps with the black image area, the relative error of the 2D-FFT is approximately four times greater and

therefore worse than the direct calculations of the YOLOv7 algorithm. Additionally, the 2D-FFT is computationally intensive, whereas the wave parameters calculated by YOLOv7 are predicted directly without additional computations.

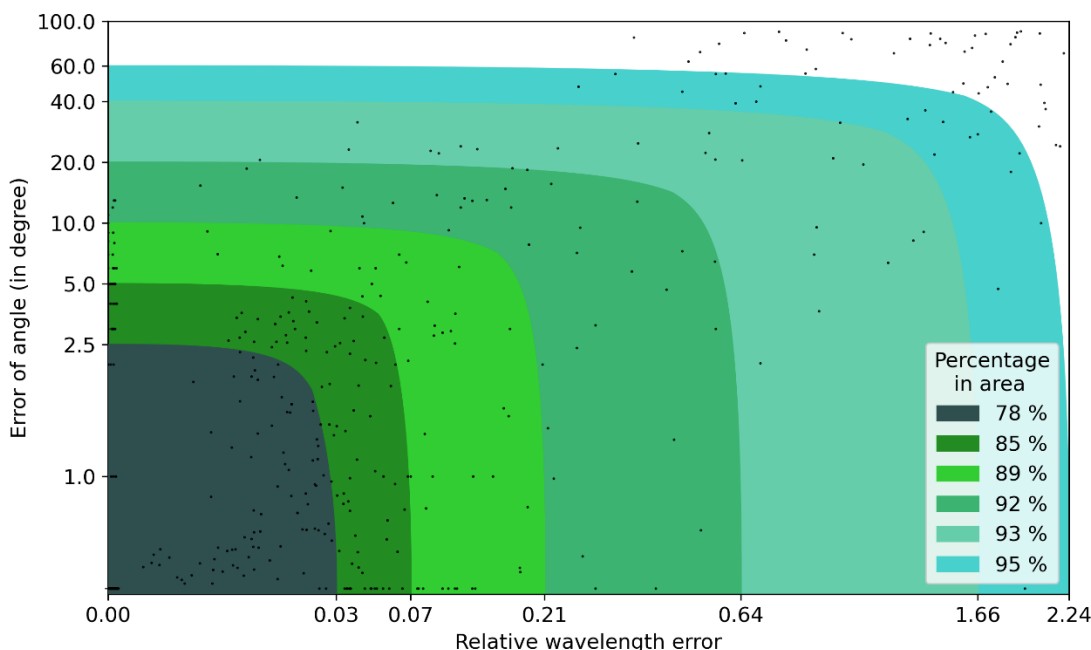

**Figure 6 Difference between labelled wave parameters and results of the 2D-FFT. The labelled parameters are regarded as correct values and the difference to the values of the 2D-FFT as an error. The values entered for the abscissa were selected so that the same number of data points have an angular error smaller than the ordinate value of the drawn circle and the relative wavelength error on the ordinate.**

## 3.3    Extraction of wave propagation direction

The algorithm presented aims to determine the direction of the propagation of wave-like structures. The relatively low temporal resolution of FAIM 4, which is approximately 123 seconds, makes it difficult to track the structures across successive images. A new algorithm is therefore developed.

This algorithm identifies the bounding boxes in sequential images which include the same wave-like structure. The decision whether two objects belong to the same wave-like structure is made as shown in Figure 7. The process begins by inserting the detected bounding boxes or a given observation period into a function, which can assess data on position, time, confidence level, and calculated wave parameters. The algorithm selects the box with the highest confidence level that has to be larger than the confidence threshold of 0.7 as the main event. It then searches for other boxes in preceding and subsequent images that have a confidence value greater than the minimum confidence threshold of 0.5 and meet specified physical proximity criteria relative to the primary event, ensuring the detected movements are within reasonable limits of wave propagation speed



and temporal lifespan. Finally, the similarity of aspect ratios between detections is assessed using the Complete IoU, CIoU (Zheng et al., 2019), to evaluate the compatibility of detected boxes. Boxes that do not meet the criteria mentioned in Figure 7 are discarded. The remaining detections are used to calculate a linear regression line that represents the best fit for the movement of the centroid of the box over time. The final result is denoted by the term "wave event" in the following. It has a defined propagation direction. This event is then saved and the algorithm iterates over the data set until no further detections

meet the initial confidence threshold.

This algorithm is applied to about 16,000 detections (with a confidence value $>= 0.5$) and leads to 1,542 wave events, each one containing at least three detections.

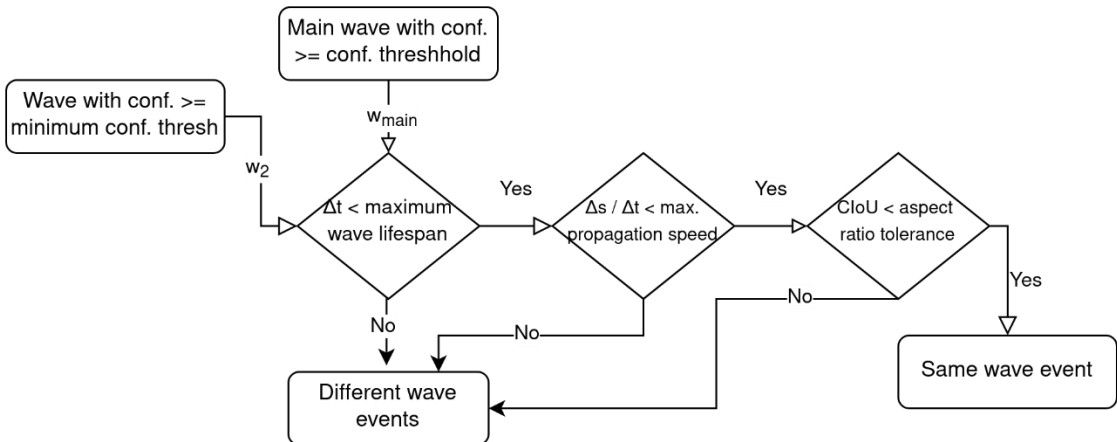

**Figure 7 Several parameters are used, including confidence threshold (0.7), minimum confidence (0.5), minimum number of detections (3), aspect ratio error tolerance (0.1), maximum propagation speed (140 m/s), maximum lifespan (20 min), and Intersection over Union (IoU) regression threshold (0.45). After applying these criteria, a best fit straight line is calculated through all the box centres of the wave event to estimate the propagation direction. Confidence is abbreviated with conf. and wave-like structure with wave.**




## 4 Results

As mentioned above, the focus of this manuscript is on small-scale wave-like structures with horizontal wavelengths in the range of ripples. This section is divided into two parts: the first one deals with the frequency of occurrence of such structures, the second one with their direction of propagation. Some results are separated according to the different seasons. If a distinction is only made between summer and winter, then the term summer refers to the months of April to September. October to March are assigned to winter. If necessary, an additional differentiation is made between spring and autumn. In this case, spring includes March until May, summer June until August, autumn September until November, and winter December until February.

A total of 264,824 FAIM 4 images taken in 941 nights between October 2020 and September 2023 are investigated, of which 49.2% are from the summer and 50.8% from the winter season. During 133 nights in this time period, the measurements were not possible due to technical problems. So, the instrument worked properly in ca. 88% of all cases. For all analyses, we use only images, whose cloud coverage probability, as it was described in section 2, is less than 10%. The number of detections of wave-like structures varies with the chosen confidence value. Starting with ca. 70,000 detections for a confidence value of 0.1, they drop sharply to less than 50% for higher confidence values (Figure 8). We choose a confidence value of 0.5. That means ca. 16,000 small-scale structures with horizontal wavelengths between 3 and 19 km are identified with a reasonable amount of confidence for our investigations. The distribution of these horizontal wavelengths is right skewed for both seasons and shows a maximum at 7 km for summer and 8 km for winter (Figure 9). The number of detections is higher in summer than in winter.




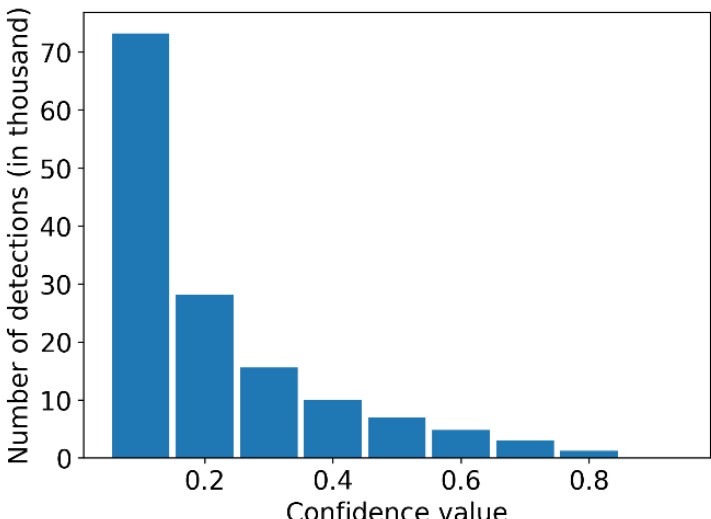

**Figure 8 The number of detections varies strongly with the confidence value: the higher the value, the lower the number of detections. A sharp decrease is observed from a confidence value of 0.1 to 0.2. It reduces the number of detections by more than 50%.**

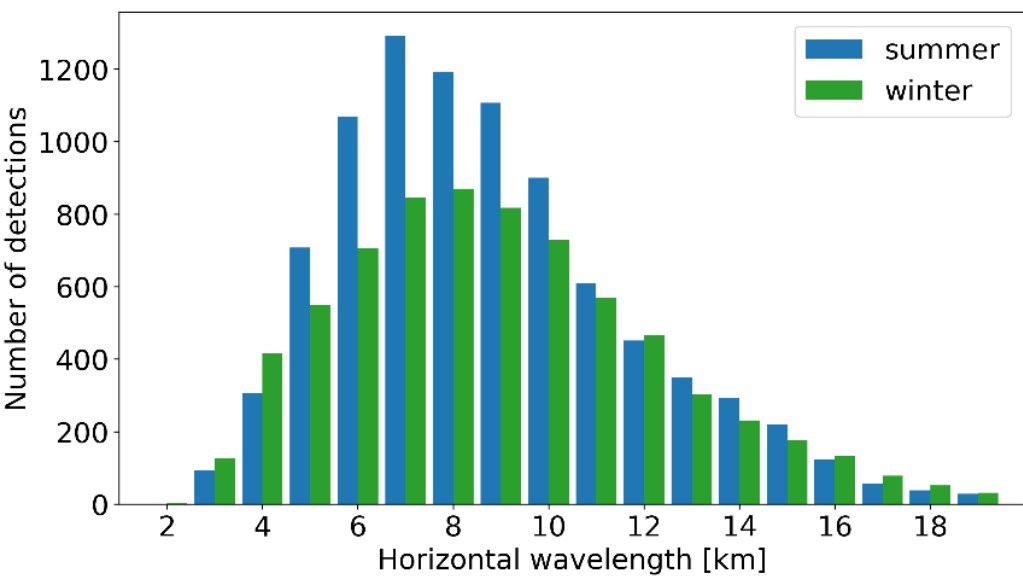

**Figure 9  The number of detections by the YOLOv7 algorithm (with a confidence score ≥ 0.5) by wavelength is right-skewed for summer (April to September) and winter (October to February). No distinction is made whether the wave is observed continuously over a longer period or only for a short time.**



If we no longer differentiate by horizontal wavelength, but calculate a kind of probability of occurrence, so add up the images with at least one small-scale wave-like structure and divide by the total number of images per season, seasonal variations

become apparent (Figure 10). The occurrence probability ranges between ca. 13% and 24% over all seasons with a clear maximum in summer, while the values for the other seasons do not differ all too much, taking their standard deviation into account. Seasons of one year with less than 900 cloudless images are excluded. This holds for winter 2020, summer 2021, spring 2023 and autumn 2023. The lower values during these seasons are a result of technical problems of the instrument and bad weather.

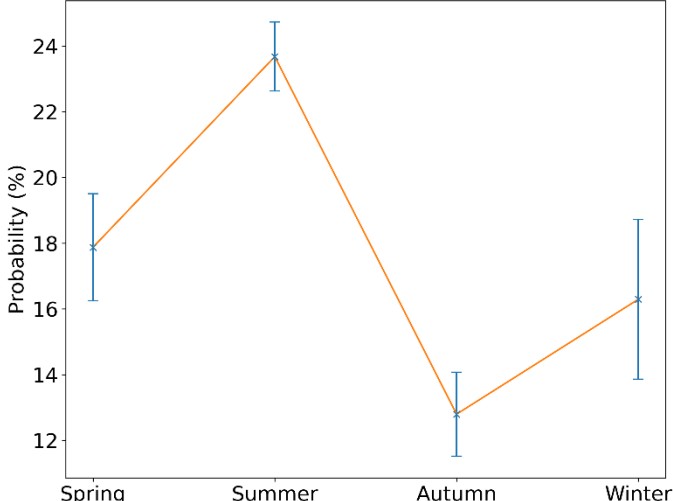


**Figure 10 Occurrence of small-scale wave-like structures in the airglow layer above Oberpfaffenhofen including their standard deviation.**

The analysis of the propagation direction can be done based on two approaches: firstly, the approach described in section 3.3

is used, i.e. in this case the direction of movement of the rectangle covering the wave structure (bounding box) is tracked. This approach is based on the analysis of FAIM videos. For the second approach, the orientation of the wave fronts is determined based on the horizontal wave parameters from individual images. Gravity waves propagate perpendicular to the orientation of their wave fronts, so 90° needs to be added to the wave orientation to give the propagation direction with a 180° ambiguity. The ambiguity is due to the use of individual images instead of a sequence of images: it is not possible to distinguish whether a north-south oriented wave is propagating eastwards or westwards, for example. If all the observed wave-like structures are

(secondary) gravity waves, then there is no difference between the propagation directions derived from the two different approaches. For propagation direction based on the individual images, there is a clear north-easterly (± 180°) component at all times of the year (Figure 11). This is most evident in summer. During the other seasons there are additional weaker maxima





with a southerly (± 180°) component. The propagation direction derived from the videos shows more variation (Figure 12).

While a west-southwest component dominates in spring, the propagation direction seems to shift to the south in summer. In autumn and winter a clear easterly component can be observed.

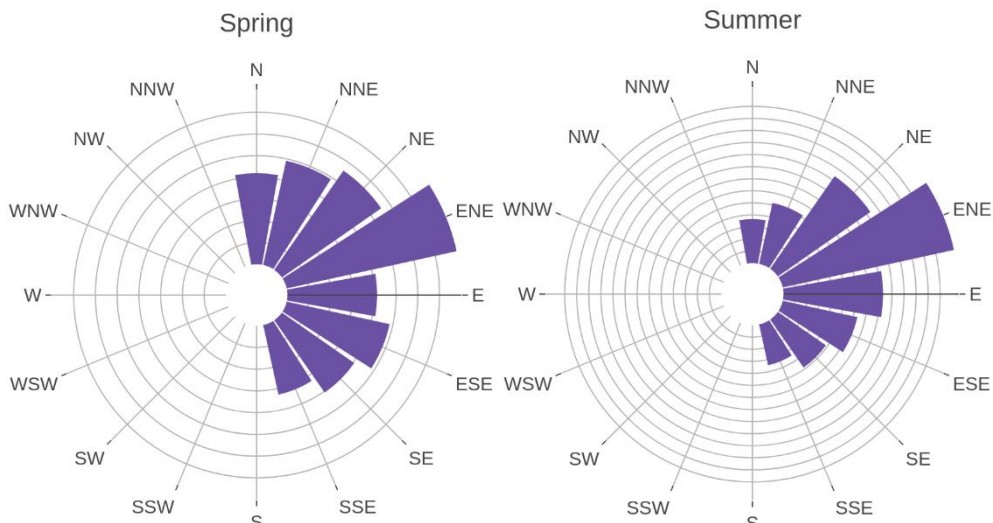


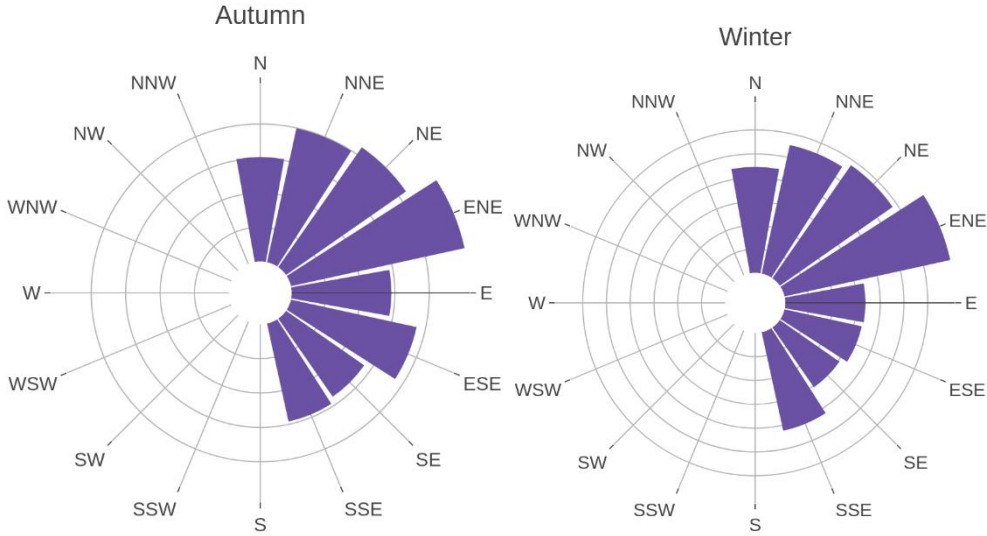

**Figure 11 Shown is the propagation direction (which is assumed to be perpendicular to the wave fronts) of small-scale wave structures in the OH\* airglow layer for all seasons, detected by the modified YOLOv7 algorithm, so based on the orientation of the**
**wave-like structure. All 16,037 detections between October 2020 and September 2023, which have a confidence score ≥ 0.5, are used. Each grey circle stands for 100 additional detections.**





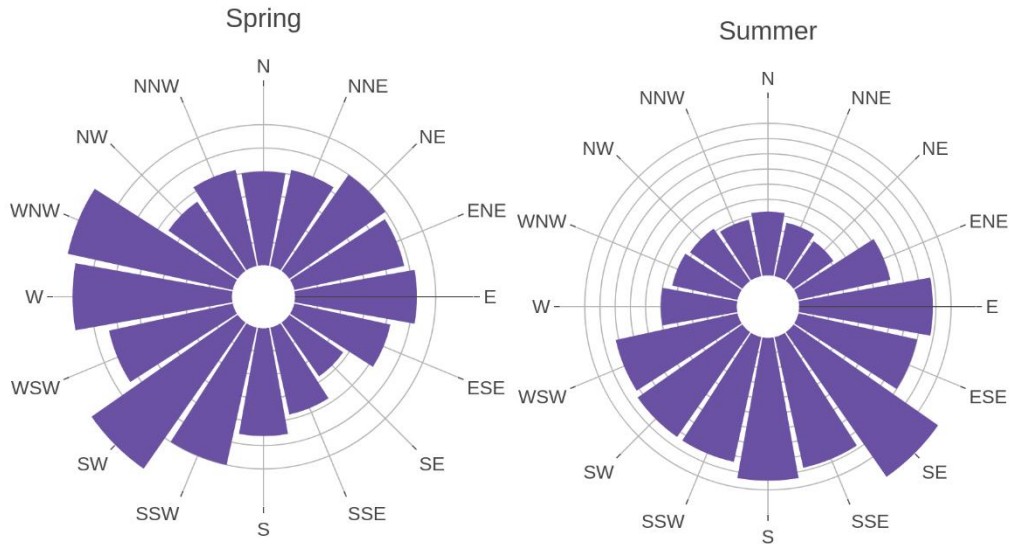


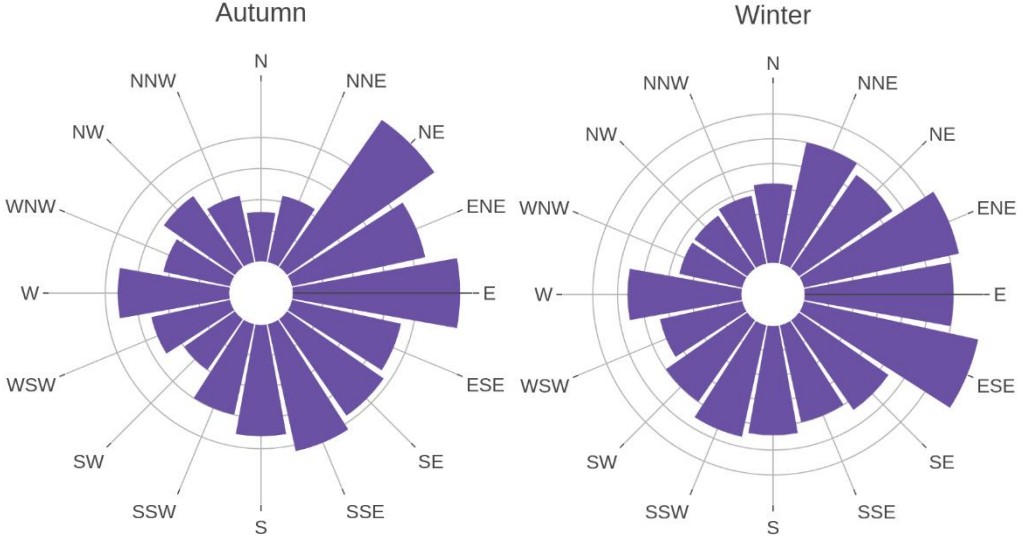

**Figure 12 Using the algorithm described in section 3.3 for all 16,037 detections with confidence ≥ 0.5, the propagation directions of the wave events are calculated. Out of a total of 1542 events, 397 occurred in spring, 531 in summer, 256 in autumn, and 349 in winter. Each grey circle stands for 5 additional detections.**




## 5 Discussion

In the following, we compare our results to other already published ones referring to ripples. As the extraction of ripples is time-consuming, there do not exist many publications that cover data of some years. We focus here on two publications referring to the Northern hemisphere (Yue et al., 2010; Li et al., 2017) and two referring to the Southern hemisphere (Suzuki et al., 2011; Rourke et al., 2017).

Yue et al. (2010) analyse data from two imagers, one at the Yucca Ridge Field Station, YRFS (40.7° N, 105° W), Fort Collins, Colorado, USA and another at two different locations in Japan (first Shigaraki, MU Observatory (34.9° N, 136.1° E) and then Misato Observatory (34.1° N,135.4° E), Wakayama, Japan). The data from the US device covers five years (2003–2008), that from the Japanese four years (1999–2003). Both systems measure OH*, but unlike ours, not above 1 μm. This means that the signal comes from a slightly different altitude range (order of magnitude: some hundred metres) than that of FAIM 4; however, this should be irrelevant for further discussion. Both instruments are all-sky imagers with a $512 \times 512$ px² sensor, the spatial resolution is not explicitly stated, but is estimated to be less than ours (see Figure 2). The temporal resolution is 2 min for the US imager which is comparable to ours and 5.5 min for the Japanese imager. The authors calculate difference images in order to emphasize small-scale travelling structures and get rid of stationary ones. This means that periods of 4 min are amplified in the data from the US imager, while periods of 11 min are amplified in the Japanese data. Periods longer than 4 or 11 min are attenuated. Some attenuation of ripples cannot be excluded, as they move with the background wind. The calculation of difference images leads to a varying filter function depending on the background wind, the horizontal wavelength of the ripples and their orientation to the wind[2]. The data is analysed manually (this is not explicitly stated, however, an analysis technique is not provided) extracting horizontal wavelengths shorter than 15 km, which are visible for a maximum of 45 minutes and cover a small geographical area ($< 5 \times 10^3$ km², this is of the same order of magnitude as our signals). In order to derive the occurrence frequency of ripples, the data is split into bins of 30 min. If at least one ripple event is identified in such a 30 min bin, it is noted down. It is not distinguished between the number of ripple events in one 30 min bin. The occurrence frequency of ripples is then calculated by dividing the number $p$ of 30 min bins with one or more ripple events by the total number $n$ of cloudless 30 min bins. The confidence is given by $\sqrt{\frac{p(1-p)}{n}}$. The data set comprises 5000 h (US) and 800 h (Japan). Together with the occurrence frequency it can be concluded that at least approximately 600 (US) and 40 (Japan) ripple events are observed. The authors find that ripples are more likely around the solstices at all three stations and therefore less likely during equinoxes. They attribute this finding to the seasonal change of gravity wave activity in the mesosphere / lower thermosphere

---

[2] Destructive interference appears for example, if the wind component which is perpendicular to the orientation of the wave is in the range of half a horizontal wavelength divided by 2 min or 5.5 min. For a wave with 10 km horizontal wavelength measured by the imager in Japan this is the case when wind perpendicular to the wave orientation accounts for ca. 15 m/s. The smaller the horizontal wavelength, the weaker the wind needs to be for the filtering to occur.





(MLT). However, they do not see a correlation between the occurrence frequency of ripples and the probability of instability
(derived from a nearby lidar) and discuss several reasons for this.

It appears that Li et al. (2017) partly use the same observations at the YRFS as Yue et al. (2010). However, they cover the
shorter period of 2003 until 2005 and use additional measurements from a sodium lidar and an MF radar. They investigate an
area of 200 × 200 km² and compute difference images. In contrast to Yue et al. (2010), they make a selection of ripples
depending on the availability of lidar and radar measurements. Thus, they analyse about 320 ripples. They define the
probability of occurrence in another way than Yue et al. (2010): Li et al. (2017) divide the total ripples' lifetime to the total
observed time under clear sky and no-moon conditions. They find a maximum of ripples in autumn (September–November)
and a minimum in summer (June–August). This roughly compares with Yue et al. (2010). Higher values in autumn and winter
and lower values in spring and summer can be read from their Figure 2b, which shows the frequency of occurrence on a
monthly basis. However, the maximum is probably in winter and the minimum in summer. This comparison may suffer from
the slightly different definition of the parameter probability of occurrence and also from the calculation of seasonal values
from monthly ones. Li et al. (2017) summarize the seasonal variation of the propagation direction of the ripples with northward
in spring and summer and southward in winter.

The analysis of Suzuki et al. (2011) and Rourke et al. (2017) both refer to measurements at or near the South Pole. The
measurement setup and the data analysis of Suzuki et al. (2011) is quite similar to that of Yue et al. (2010). The authors use an
all-sky camera (512 × 512 px²) and calculate time difference images. The temporal resolution is slightly better than the one of
our scan images with about 100 s and they observe a slightly higher airglow layer (sodium, ca. 90 km height). Due to polar
night conditions, their data cover the months April–August of the years 2003–2005 and they observe 213 cases of ripples. For
them, ripples have horizontal wavelengths shorter than 17 km. They observe propagation directions towards 90°–120°E and
300°–330°E.
Rourke et al. (2017) refer to wavelengths shorter than 15 km as ripples. In contrast to the other studies mentioned above, they
do not use an all-sky imager but a scanning radiometer (UWOSCR, University of Western Ontario SCanning Radiometer),
which is based on an InGaAs photodiode. The UWOSCR is sensitive to the spectral range between 1100 and 1650 nm and
provides information about an area of the night sky of 24 × 24 km² divided into 16 × 16 px². The authors use a Fast Fourier
transform (FFT) to determine the wave period between 2 and 16 min and correlation analyses in orthogonal directions of the
image to derive the zonal and meridional velocity components. Horizontal wavelengths are calculated from the period and the
horizontal phase velocity. Ripples are mainly observed from March to June and very few ones from July to October. Most of
them propagate westward and poleward and have a horizontal wavelength of 8–10 km. This analysis covers the years 1999 to
2013 and includes about 400 cases.



Compared to the results of the authors above, ours are based on about two orders of magnitude more small-scale wave-like structures, although the number of observation years is of the same order of magnitude. Since Oberpfaffenhofen, which is North of the Alps and in the densely populated area of Munich, is not a station with outstanding good observation conditions

in terms of cloud coverage and artificial illumination, this effect can be attributed primarily to the use of the developed machine learning algorithm. Other reasons are the better spatial resolution compared with all-sky imagers and possibly also the use of original images instead of difference images. The latter can lead to a reduction of the data basis.

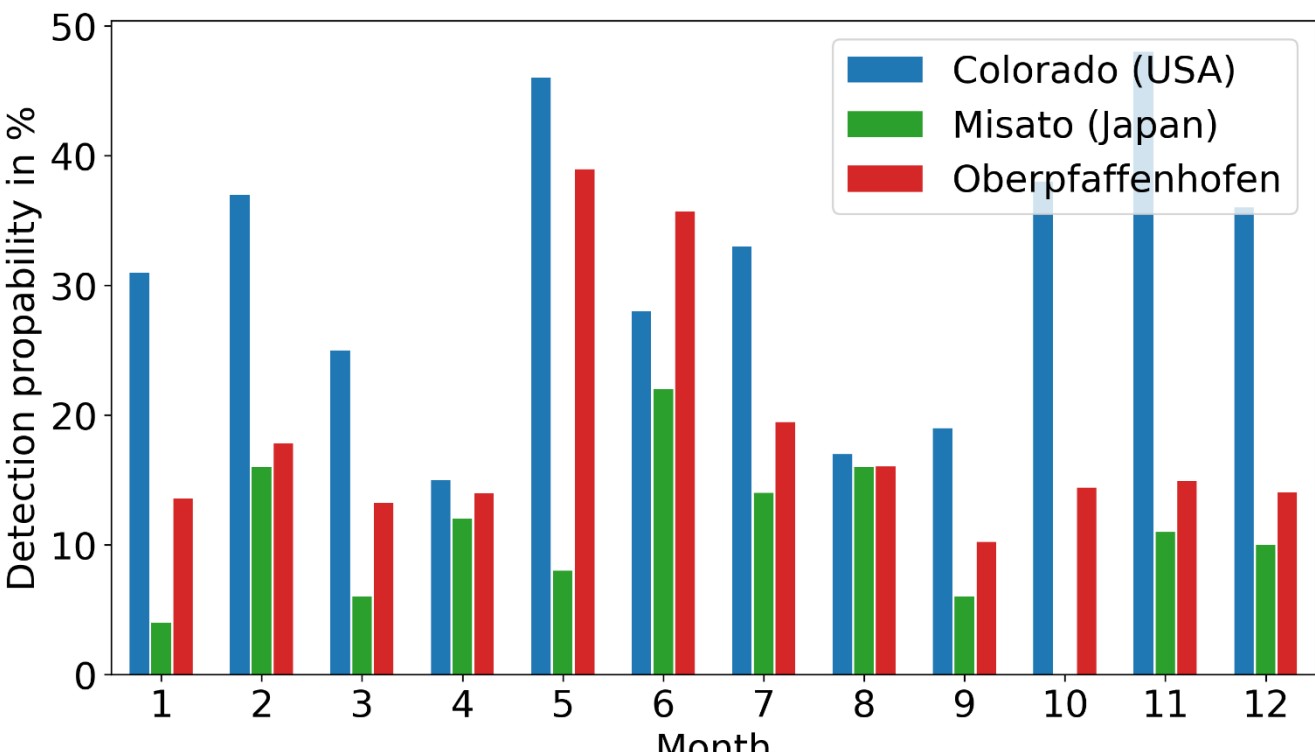

**Figure 13 Average detection probability for the Japanese and American measuring devices according to the data from Yue et al. (2010) compared to our results in Oberpfaffenhofen for the different seasons.**

It is not possible to compare the absolute probability or frequency of occurrence, as the various authors define these slightly differently. However, one can compare the relative course. Figure 13 shows the values read from Yue et al. (2010) and our

own. The values for the southern hemisphere are not included here, as these authors either do not provide them or it is not possible to read them from their figures. Even if there are differences between the individual courses, the data sets show increased values from May or June to July or August. Additionally, the US observations are characterized by significantly higher values from October to February. This does not hold for the Japanese data and our own. When mentioning the period



May to July or August, one should note that this is the time when the OH* measurements actually address the mesopause. The rest of the year, the mesopause is higher (at approx. 100 km, She et al. (2000), Berger and Zahn (1999)). The temperature gradient in the mesosphere is thus steeper from May to August, which leads to an increased probability of convective instability. As far as dynamic stability is concerned, one can read in Jacobi et al. (2015) who measured winds at Collm, Germany, which is approximately 380 km northeast of Oberpfaffenhofen, that the meridional wind reaches its maximal absolute value in April and/or May, the zonal wind in August. These are long-term mean values, which can of course vary for individual years. The mean and the median height of the signal is between 89 km and 90 km, so relatively close to the OH*-airglow layer height. Therefore, we conclude that the probability for dynamic instability is enhanced for the OH*-airglow layer height in the summer season. This increased tendency to instability, convective or dynamic, could therefore contribute to or explain the higher probability of instability features (showing up as small-scale wave-like signatures) at this time of year. Even if the instability tendency remains constant, a higher probability of such instability features may also occur during May to July/August, if more gravity waves, which might break, are observed at this time. This is indeed the case as Figure 14 shows. The number of wave events in this figure with horizontal wavelengths longer than 15 km are derived according to Hannawald et al. (2016) from non-scanning FAIM data sets recorded in Oberpfaffenhofen and some neighbouring Alpine regions (SBO: Sonnblick Observatory (47.05° N, 12.95° E), Austria and OTL: Otlica Observatory (45.93° N, 13.91° E), Slovenia). They are characterized by a maximum in spring, especially in May. The smaller-scale instability structures with wavelengths shorter than 15 km have a maximum in the summer months May to July. Similar results were obtained by Wüst et al. (2016) and Sedlak et al. (2021), however, these authors do not focus on the number of waves, this parameter contributes indirectly to their results. Based on OH*-airglow spectrometer data of different European stations (in the vicinity of the Alps), Wüst et al. (2016) show that gravity waves with periods greater than 60 min transport more potential energy not only in winter, but possibly also in May and June. Sedlak et al. (2020) who evaluated FAIM data in the Alpine region with regard to their gravity wave activity and depending on their period, see a similar behaviour for medium-frequency waves. Low-frequency waves have a maximum in winter, high-frequency waves are characterized by a relatively constant activity. This could be interpreted as an indication that the observed small-scale wave-like structures are due to medium-frequency waves.



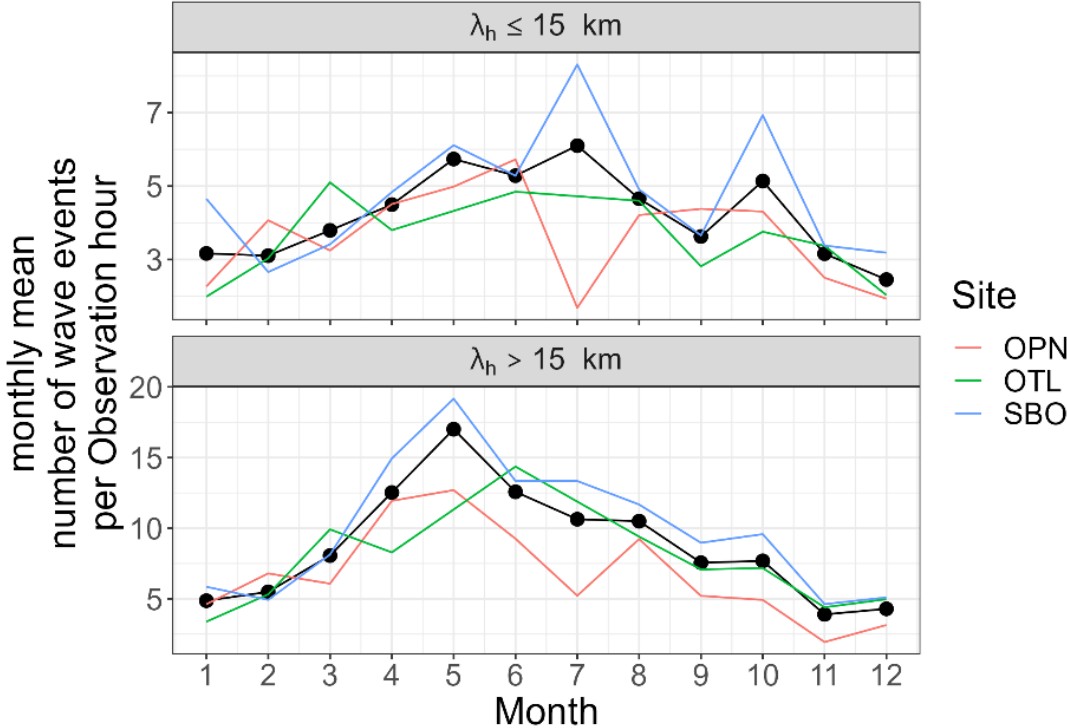

**Figure 14** Shown is the (averaged) number of GW events during the year based on data from the narrow field airglow imager
FAIM 1 for different stations (SBO: Sonnblick Observatory, Austria, OTL: Otlica Observatory, Slovenia, OPN: Oberpfaffenhofen,
Germany) in the Alpine region or its vicinity. FAIM 1 has a high temporal resolution of 0.5 s per image and observed OH* airglow
during different years at the different sites: the OTL data set is analysed with the 2D-FFT (see Hannawald et al. (2019) for the
algorithm pipeline) from October 2017 to October 2018, OPN from December 2013 to January 2015 and SBO from August 2015 to
July 2017. Wavelike structures with horizontal wavelengths larger 15 km have a maximum in May. The very small-scale wave-like
structures are generally more frequent during the summer months in the Northern hemisphere.

Due to the work of Li et al. (2017) at the latest, however, it is clear that such small-scale structures could equally be secondary

gravity waves. These waves are generated through breaking primary GWs (see e.g., Becker and Vadas, 2018 and citations

therein). In our measurements, as in many others, it is not possible to distinguish between secondary and primary waves.

Further discussion at this point is therefore superfluous.

However, we can use the fact that (secondary) gravity waves propagate perpendicular to the orientation of their wave fronts

and instabilities move with the wind to estimate at least the season when our small-scale wave-like signatures are more likely

to be instability features than secondary waves. The direction of propagation shown in Figure 11 (which is ambiguous for

180°) is based on the assumption that the observed wave-like structures propagate perpendicular to their direction of

orientation. It is retrieved from individual images. The propagation direction, which is shown in Figure 12, is derived from the



propagation direction of the rectangles enveloping the wave-like structures in the videos. In order to compare the results, the propagation directions from Figure 12 are artificially degraded, i.e. 180° are added to all propagation directions with a

westward component so that they also lie in the range between north and south-southeast (Figure 15). Since a wave-like structure must be present in several images in succession in order to capture its direction of propagation in the video, the number of observed directions of propagation is lower compared to the one retrieved from individual images. The results are therefore normalized for better comparability. Due to the different amount of data, the results are not compared quantitatively. Qualitatively, however, the difference between the propagation directions is most pronounced in summer. At this time of year,

the identified wave-like structures therefore most clearly do not propagate perpendicular to their fronts. This suggests that instability signatures are observed above Oberpfaffenhofen especially in summer and that the interpretation of ripples as instability structures is most likely to be correct at this time of the year.



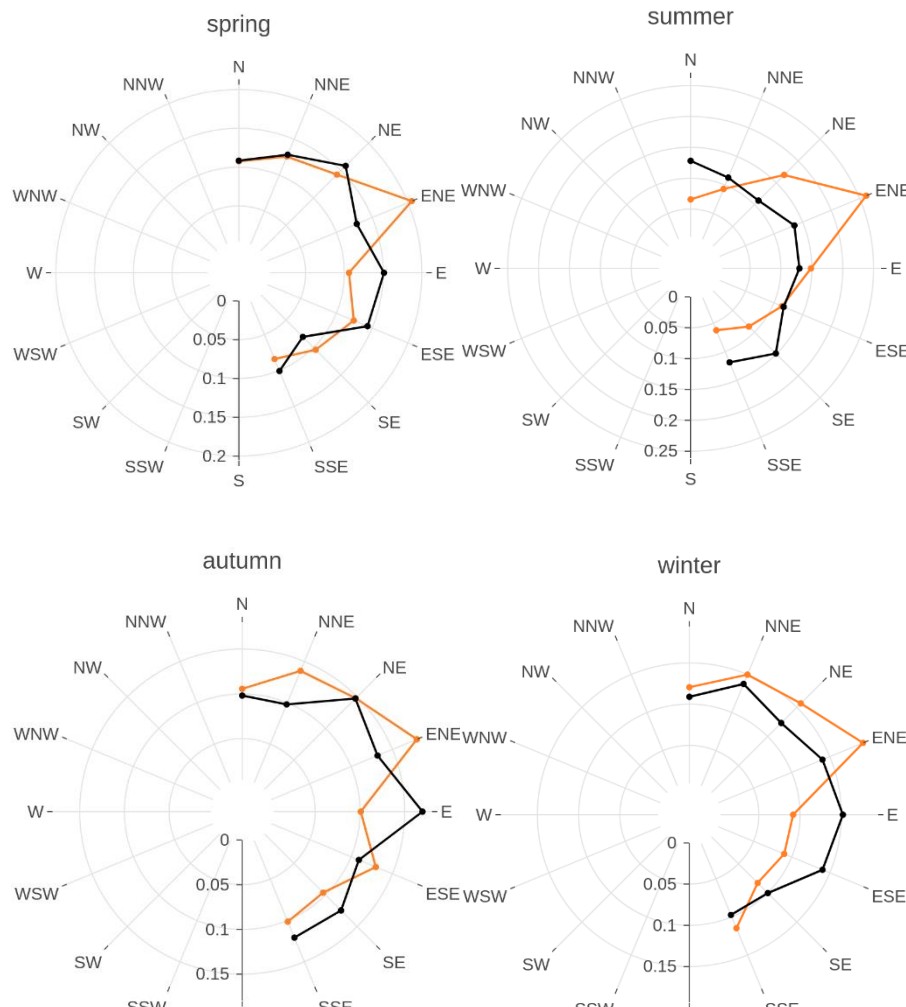


**Figure 15 Shown is the comparison of the wave propagation directions (including an ambiguity of 180°) derived with the two different algorithms. The values plotted in orange refer to the individual frames. Here, it is assumed that the wave-like signatures propagate perpendicular to their fronts. The propagation directions derived from the movement of the bounding box (see Figure 12) are plotted in black.**


Concerning the direction of propagation, we observe in our data changes from south in spring to east in winter (with variations to the south-east and north-east in summer and autumn). In general, all propagation directions occur at all times of the year. Li et al. (2017) also observe all directions of propagation except in summer. In spring, their analyses show a broad southerly component as well as a clear peak to the North. This peak persists in summer. It can no longer be observed in autumn and

winter. In autumn, it is in general difficult to determine a dominant direction. A slight preference for a meridional direction may be observed. In winter, a minimum towards the west and a southern rather than a northern component can be recognised. In contrast to Li et al. (2017), we do not observe this strong northern component at any time of the year. However, we agree



in the broad maximum to the south in spring. In winter, we observe a preferred eastward component, while Li et al. (2017) derive a minimum in westward direction. This is at least not contradictory. However, in summer, our results and those of Li et

al. (2017) are contradictory and while the results of Li et al. (2017) are less clear in autumn, we observe a clear eastward and poleward component. In the southern hemispheric autumn (March to June), a strong westerly and poleward propagation direction is observed (Rourke et al., 2017). A comparison with Suzuki et al. (2011) is not made here, as the station is located directly at the South Pole and is therefore subject to specific meteorological conditions.

So, there appears to be less agreement with regard to the direction of propagation than with the seasonal frequency of

occurrence. One reason for this could be that local effects dominate. If the wave-like structures are interpreted in terms of instability features, one could argue that they move with the local wind and the wind is variable in space and in time, even during one night since it shows strong variations due to tidal activity. As already mentioned above, the interpretation as instability features is most likely correct in summer and indeed in this season, our propagation direction and those of Li et al. (2017) are contradictory.




## 6    Summary and outlook

In this study, we analyse ca. three years of OH* airglow all-sky images for spatially confined wave-like structures with horizontal wavelengths of ca. 20 km and smaller. The data were recorded by a scanning OH* airglow camera, which is operated operationally every night at DLR Oberpfaffenhofen (48.09°N, 11.28°E), Germany since 2019. It provides nearly all-sky
images (diameter ca. 500 km) with a spatial resolution of ca. 150 m/px (at 30° zenith angle) and a temporal resolution of ca. 2 min. In order to identify those small-scale and spatially confined structures, we adapt and train YOLOv7, a deep convolutional neural network, to derive the position and the horizontal wavelength. Our results can be summarized as follows:

1.   We identify ca. 16,000 small-scale wave-like structures. Studies mentioned in literature cover two orders of magnitude fewer events. The main reason is that former studies are based on the visual inspection instead of machine
605        learning. Further reasons might be that we use a scanning imager that provides a better spatial resolution compared to non-scanning all-sky imagers. Scanning photodiodes cover a smaller part of the sky or work with lower resolutions than we do. Our mode of operation therefore increases the probability of observing small-scale wave-like structures.

2.   As far as the seasonal frequency of occurrence is concerned, our results differ to some extent from those of the other authors who analysed their data manually and used mostly difference images. However, all data sets show increased
610        values from May or June to July or August. Additionally, the US observations of Yue et al. (2010) are characterized by significantly higher values from October to February. The summer maximum can have different reasons. It could be due to an increased probability of convective and dynamic instability but also to a larger amount of gravity waves during this time of the year. All reasons are supported by literature.

3.   Concerning the direction of propagation, we observe in our data all propagation directions at all times of the year.
615        However, the preferred direction changes from south in spring to east in winter. There appears to be less agreement between our results and literature values with regard to the direction of propagation than with the seasonal frequency of occurrence. One reason for this could be that such small-scale wave-like structures can be due to instabilities. Those instabilities, often called ripples, move with the wind and the wind is variable in time, even during one night since it shows strong variations due to tidal activity, and space.

4.   However, those small-scale wave-like structures do not need be due to instability, they can also be secondary waves. With our measurements, we can only indirectly differentiate between those two features. Based on the direction of propagation, we deduce that instability signatures are observed above Oberpfaffenhofen especially in summer and that the interpretation of ripples as instability structures is most likely to be correct at this time of the year.

In order to obtain more precise information about the reason for the occurrence of these small-scale structures in future,
the scanning FAIM system is to be combined with a camera with better spatial resolution. This will make it possible to visualise turbulent structures and thus not only distinguish between instabilities and secondary waves but also derive information about the amount of energy transferred (e.g. following the approach of  Sedlak et al. (2021) that needs to be adapted to large data sets). This means that the possible position of the wave-like structure is to be recorded using FAIM 4,



the position is forwarded to the higher-resolution camera, which then aligns itself with it (operating-on-demand). The

work presented in this manuscript is considered a first step in this direction.

Finally, here are some ideas on how to improve the modified YOLO algorithm: The most obvious would be the addition

of more labels. This generally leads to more robust and universal detections. A major issue in creating the labels and in

subsequent training is the subjective boundary between an event worthy of annotation and one that is not. Here, it would

be advisable to also create confidence values for each box while labelling, and to modify the YOLOv7 algorithm so that

it predicts not just zero or one for the object value, but the labelled confidence value. The same data set could also be

labelled by multiple scientists to calculate confidence values based on the intersection of labels. Another possibility would

be the labelling of "ignored areas", where the scientist is not sure about the existence of wave structures. These areas could

then be ignored by the loss function. However, each of these approaches is pretty time-consuming and may be done in the

future.




**Author contribution**

This work was funded in parts by different project (AlpEnDAC Phase 2, VoCaS-Alp and LUDWIG), which were acquired by MB and SW. VoCaS-Alp and LUDWIG were managed scientifically by SW. FAIM 4 was setup up by JoS and PH, it was operated by PH. PH also provided the basic image processing. RL advised in the tasks and solutions suitable for machine

learning on FAIM data. JaS modified the YOLOv7 structure, applied it to the FAIM 4 data and derived the results in his master thesis. SW defined the topic of the master thesis, supervised it and interpreted the results. The manuscript was written by SW, supported by JaS and PH. All authors read the manuscript.

**Acknowledgement**

The setup of FAIM 4 was funded by the Bavarian State Ministry for the Environment and Consumer Protection within the
project AlpEnDAC Phase 2 (grant number: TUS01UFS-72184) as an example for operating on demand. The purchase of the static FAIM cameras, their deployment and the analysis algorithms were funded by the same ministry within the projects LUDWIG (grant no. TUS01UFS-67093) and VoCaS-Alp (grant no. TKP01KPB-70581).

We used data from the work of Sedlak et al. (2021) and thank the authors for this.

Furthermore, we thank Ridvan Salih Kuzu (DLR), Gerhard Schauer (Sonnblick Observatory), and Samo Stanič (University of
Nova Gorica): Ridvan Salih Kuzu helped with the development of the machine learning algorithm. Gerhard Schauer and Samo Stanič maintained the FAIM 1 imager at Sonnblick Observatory, Austria, and at Otlica Observatory, Slovenia, respectively.

We did not only use artificial intelligence for the analysis of the data, this manuscript has also been linguistically improved by the help of deepl.com and chat.openai.com.

**Data availability**

The FAIM data is stored by the Network for the Detection for Mesospheric Change, NDMC, the server is hosted by DLR. They are available on request.

**Competing interest**

The authors declare that they have no conflict of interest.




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
