# Peer review of "Extraction of spatially confined small-scale waves from highresolution all-sky airglow images based on machine learning"

_EGUsphere, 2025_

## Author Comment (AC1)

This manuscript presents a method for detecting small-scale airglow wave structures using a modified YOLOv7. The paper is well written, scientifically sound, and a welcome contribution. A few concerns regarding the methodology need to be addressed, however. Below are my itemized comments.

We thank the reviewer for the very valuable comments. We answered all of them (in orange, see below) and changed the manuscript accordingly.

- Line 173. 'BYOL....'

  Pretraining using BYOL may not be particularly beneficial for this application, and the manuscript provides no evidence that BYOL improves performance. Nevertheless, the authors should at least include further details on the implementation of BYOL and YOLOv7. In particular:

  1. **Which variant of YOLOv7 was used? YOLOv7 has multiple variants with different backbones and capacities.**

     Yolov7-tiny (info added in the manuscript)

  2. **Was the BYOL pretraining performed starting from a randomly initialized network, or from an already pretrained YOLOv7 backbone (for example COCO pretrained)?**

     It was trained from a COCO pretrained network, which is provided by the official paper (info added in the manuscript).

  3. **What data augmentations were used for BYOL during pretraining?**

     Colour Jitter, Random Flips, Random Resized Crop, Normalization, Gaussian Blur (info added in the manuscript)

  4. **Did the authors compare the performance of the BYOL-pretrained model with a non-pretrained (trained from scratch) or standard pretrained YOLOv7 model?**

     Yes, compared to the regular COCO pretrained model. While the training accuracy was not significantly better, the convergence speed was increased.

- YOLOv7 is adapted to output wavelength and orientation. This is in fact a substantial change to the net and these tasks deviate from the original design goal of the YOLO structure. YOLO and its variants are highly optimized for object detection but not much for extracting information from the objects. Adding three additional regression features may severely interfere with its main task (object detection). Generally speaking, regression tasks in neural networks require careful design of the architecture, loss functions, and training strategy. Simply adding three additional regression outputs to the YOLOv7 head is unlikely to work well without further justification or validation.

  **Has the author tried using YOLOv7 in its original form and compared the numbers?**

  Yes, and the accuracy does not change when no additional parameters are used. However, in this case, it should also be mentioned that the value ranges for the newly generated predictions have been carefully selected.

The wavelength, for example, is calculated in relation to the bounding box size. Since the wavelength must be shorter than the box size, the value range remains exactly the same. Both predicted values (bounding box and wavelength) are initially in the range $(-\infty,\infty)(-\infty,\infty)$ and are then mapped to a range of $(0,4)(0,4)$ using the same formula:
$(2\cdot\mathrm{sigmoid}(x))^2$
This was done to ensure that the gradients do not differ too much in magnitude and that both bounding boxes and wave parameters are predicted with high precision.

- **Line 284. A validation set is absolutely necessary. The testing set is the one that is optional. In recent work, some studies omit a separate testing set and report validation metrics only, provided that the validation set is sufficiently large and representative. Omitting the validation set, however, is not consistent with standard neural network training practice, since it prevents proper monitoring of overfitting and reliable model selection. Without a validation set, it is impossible to detect overfitting during training. Given that the training set is relatively small (only in the thousands), not using a validation set is a fatal mistake, and the performance metrics obtained during training are likely to reflect overfitting rather than true generalization.**

  We believe this might simply be a naming issue. All model architectures and hyperparameters were fixed *a priori*, and the test dataset was used only for evaluation. There was no adjustment of hyperparameters during training, nor were any other training decisions made based on that dataset. Nevertheless, we repeated the training while splitting the test set into separate test and validation sets of equal size. No overfitting was detected in this setup, and the precision and recall values remained unchanged.

- Line 288. '......78% are correctly identified'.

  **This is not an appropriate way to report regression performance. Regression tasks should be evaluated using continuous error metrics such as MSE or RMSE, and wavelength and orientation should be reported separately with their respective error distributions. Using a binary threshold to count predictions as "correct" obscures the actual performance and does not provide enough information to assess model accuracy.**

  The decision to use a binary classification between correct and not correct was done, since metrics like MSE or RMSE would hide this important information. For us, the required minimal level of quality was important to get an estimate of how many predictions are "correct" and therefore how trustworthy the calculated propagation directions (in the later chapter) would be.

- Figure 5 and ~ Line 286.

  **The reported performance is subpar for a task that should not be particularly difficult for a modern neural network. This suggests there might be issues with the data, the net config, and/or training. I suggest that the authors retrain the network without the additional regression features, expand the training data if possible, and include a validation set. If the size of the training dataset is the main constraint, using the testing set as the validation set and reporting the validation metrics is also acceptable.**

**The orientation and wavelength can be handled much more effectively by a dedicated CNN or ViT that processes the image content within the bounding box. Or even better, a DETR-based model would be more suitable for predicting both the bounding box and the orientation. However, adapting the method to DETR would require substantial additional work and is not strictly necessary here.**

Thank you for this valuable feedback.

Another training with validation and test set was performed as mentioned above. No overfitting was detected, and the precision and recall values remained unchanged. For the additionally calculated metrics, please have a look at the answer below. Furthermore, we used YOLOv7 also in its original form (without using additional parameters), as written above, and the accuracy did not change.

As mentioned in the manuscript, one inherent problem is the ambiguity of wave events. There were many cases where wave events were on the cusp of being classified as such. Therefore, the model might predict a wave that was not labelled as such, and then be trained not to detect it as a wave anymore, and vice versa. The best way to avoid this issue is to label the probabilities of wave events and adjust the training accordingly. In the current solution, a '50 % wave event is trained to be predicted with either 100 or 0 % confidence, which is 'half wrong' either way.

Expanding the data set would of course be beneficial. However, the project in which this work was performed is finished and Jakob Strutz, who did the AI analysis is working in another job. Therefore, enlarging the data set us not possible.

**Line 320. 'In summary, the 2D-FFT provides more accurate results as 78% of the FFT predictions have an error of less than or equal to 2.5° for the orientation and 3% (relative to the labelled wavelength) for the wavelength. For the modified YOLOv7 algorithm, 78% of the results are considered correct, if the wavelength error is less than 10% relative to the labelled wavelength and the error of the angle is less than 10°.'**

**This is not a fair comparison. The 2D-FFT results are evaluated using an error threshold of 2.5° for orientation and 3 percent for wavelength, while the YOLOv7 results are evaluated using a much looser threshold of 10° and 10 percent. Because the criteria differ by a large factor, the "78 percent correct" numbers for the two methods cannot be directly compared.**

While I understand the authors are trying to show that 2D-FFT performs better on normal images, the comparison is still pretty weird. It would be better to compare both methods under the same benchmark. MSE or RMSE is the standard metrics for regression tasks like these.

We fully understand your point. In this section, we intended to highlight that achieving the same accuracy (78%) can be accomplished using stricter requirements for the FFT than for the direct calculation using YOLO.
In the meantime, we performed both calculations using a relative wavelength error of 20% (compared to the labeled wavelength) and an angular error of 10°. This appears reasonable, as wavelength errors of 20% and angle errors of 10° would not significantly affect the calculation of the propagation direction or the classification between secondary waves and ripples.
Using this metric, 89% of the FFT predictions are correct, with an MSE of 7.60 km² for the wavelength and 220.71(°)² for the angle. For the YOLO predictions, 84% are correct using the same metric, with an MSE of 10.02 km² for the wavelength and 390.10(°)² for the angle.

---

## Author Comment (AC2)

Comments on the manuscript 'Extraction of spatially confined small-scale waves from high resolution all-sky airglow images based on machine learning' by Sabine Wüst et al.

This paper reports the high resolution/wide area observations of OH airglow images using a scanning camera at DLR Oberpfaffenhofen, and a new method of analyzing ripple structures in the image using ML technique. The authors have also shown the statistical results of the ripples. The new analysis technique has extracted two order larger number of events than the past literatures, and the results are well compared with the past observations.

The reviewer would like to congratulate the authors' successful observation and analyses. Th e new method applied to a wide-horizontal range and high-resolution images is very capable of studying the statistics of ripple structures (small-scale wave-like structures) in the image, for which the relations with the instabilities and secondary gravity waves are of great interest.

However, there are some points that need to be improved before the manuscript is published. Thus, I would like to recommend 'minor revision'.

Thank you very much for the congratulation and also for the very valuable comments. We answered all of them (in orange, see below) and changed the manuscript accordingly.
One technical remark concerning the track changes mode: we use Citavi and somehow changes made with Citavi are not tracked in Word. We included one new reference (Jaen et al. 2023) and shifted another one (Jacobi et al., 2015). You will recognize it from the context, however, it is not marked as a change.

Main point:

**The wording of propagation direction**

There are many places where the authors mention 'propagation direction'. I understand there are three meeting what 'directions' mean.

 (1) Direction of so called 'phase velocity', which is the apparent phase velocity of phase front lines.

 (2) Direction of motion of the area that the wave-like structure 'packet' is moving to.

 (3) Direction perpendicular to the phase front line (of a single image)

My understanding from the text is that (1) is the one we normally use in case of gravity waves and 2D-FFT is showing this by its peak (with 180 ambiguities in case of a single image.) (2) and (3) can be derived from ML technique shown here. I would like to suggest the authors to use clearly different words for (2) and others, to avoid confusion of the readers. My suggestion is to

use the word like 'direction of the wave structure movement', 'direction of wave packet motion', 'direction of wave migration', 'direction of wave drift' etc. Or, Li et al. (2017) uses 'advection', instead. I would prefer not to use the word of 'propagation' for (2) because it is not related to wave parameters but observed area. I hope such wording separation would help readers to understand the paper correct.

Thank you for this very valuable hint. We went through the manuscript and replaced propagation direction with direction of advection, if we refer to case (2). We also tried to be more precise concerning the description of other publications, however, this is somehow challenging. Suzuki et al. (2011) separate their observed wave events according to the wavelength into ripples and gravity waves and provide in both case propagation directions. From their manuscript one gets the impression that this is the direction into which the structures move. So, we changed, for example, the sentence "They observe propagation direction towards …" into "These structures move mainly towards …".

Related question.

L 410-412

'If all the observed wave-like structures are (secondary) gravity waves, then there is no difference between the propagation directions derived from the two different approaches.'

I do not understand what this sentence means. Even in case of secondary gravity waves, wave packet motion direction may not be the same as the phase velocity of the wave. Please explain more.

I had a knot in my brain, thanks for bringing my attention to it.

Instability features are supposed to move with the background wind (see e.g. the argumentation in Li et al. (2017) in order to discriminate instability features from secondary gravity waves), gravity waves do not need to do this. In the intrinsic frame, they move perpendicular to their wave fronts; this can look different when we observe them while not moving with the background wind. For waves with smaller horizontal wave numbers (larger horizontal wave lengths), the difference between the intrinsic wave propagation direction and the observed wave propagation direction (direction of advection) is less pronounced than for waves with larger wave numbers (smaller wave lengths), as the intrinsic frequency differs less from the observed one. That is probably the reason why gravity waves observed in all-sky airglow imagers seem to move perpendicular to their wave fronts when observed from the ground; we focus on the large waves and not on the small ones.

Figure 15 shows the intrinsic wave propagation direction and the observed wave propagation direction (direction of advection) on a statistical basis. As the observed horizontal wave lengths should not vary to much with the seasons, the differences between both directions are driven by the background wind speed and direction relative to the intrinsic wave propagation direction. In summer, this effect is most pronounced (this is always limited by the 180° ambiguity). Summer is also the season when the highest wind horizontal velocities are observed.

For discriminating between secondary waves and instability features, the direction of advection and the direction of the background wind are compared. According to Jacobi et al. (2015), who

measured approximately 400 km away from Oberpfaffenhofen, the averaged wind is for SW (NE-ward) for spring, NW (SE-ward) for summer, and N/NE (S/SW-ward) for autumn and winter. The dominating directions of advection in our measurements (see Figure 12) is SW in spring, S in summer, NE in autumn and E in winter. So, only in summer the observed small-scale wave-like features are overall advected with the wind. Therefore, the probability that an observed small-scale wave-like structure is an instability feature (secondary gravity wave) is highest (lowest) during summer.

We added this info in the manuscript and deleted the sentence 'If all the observed wave-like structures are (secondary) gravity waves, then there is no difference between the propagation directions derived from the two different approaches.'. Furthermore, we changed the conclusion drawn from figure 15 into "Qualitatively, however, the difference between the intrinsic propagation directions and the directions of advection (observed propagation direction) is most pronounced in summer."

**Other points**

1. 40-49

The authors describe airglow imaging observation of breaking gravity waves with citation of a few papers. To my knowledge the first clear observation of showing gravity wave breaking by airglow and its analysis was published by Yamada+ (GRL, 2001, DOI: 10.1029/2000GL011945 ) and Fritts+(GRL DOI: 10.1029/2001gl013753) . I would suggest to cite these papers which are earlier publication by about 20 years.

Done

L 94-100

The description of the FAIM 4.

It would be useful if the authors can also provide chip (or pixel) size of the InGaAs camera, and F number of the lens (or the effective aperture of the lenz) for the reader to understand the sensitivity of the optics (e.g. for knowing 'A x omega' value (throughput) of the camera).

The pixel size was already given in the former line 95 (320x256 pixel), we added the F-number.

L 173-179

It would be helpful, if the authors briefly introduce how the FOVs of FAIM 4 and FAIM 3 (13 km x 13 km?) are different.

Info added.

L 214 – 215

'Firstly, performing a 2D-FFT, especially on high-resolution images, is time-consuming and computationally expensive, leading to longer processing times and significantly affecting efficiency in analysing large data sets.'

(similar expression is at L 325.)

My feeling is that 2D-FFT is not so time consuming nowadays, as long as the number of points (most efficient one is 2^N) is selected properly. I would like to know how difficult it is to use 2-D FFT for the images introduced here. I believe zero-padding to make a square image of (2^N) * (2^N) size would make the computation time short enogh.

That is true and we added here some info: Firstly, performing a 2D-FFT, especially on high-resolution images, is time-consuming and computationally expensive as not only one 2D-FFT needs to be calculated but several ones using differently sized sub-samples of each image. This is necessary since the FFT assumes that the waves are present in the whole image. If this is not the case – which is very likely when sensing the whole sky – their amplitude is underestimated. This leads to long processing times and significantly affects efficiency in analysing large data sets.

L 517 – 522

The authors refer to Jacobi et al. (2015) and speculated that the meridional wind is strong in April/May, and the zonal wind is in August, which can explain the probability of dynamical instability is large. I do not understand this logic. Why the largest wind at around OH altitude shows probability of dynamical instability, without a measurement of wind shear.

You are right, the argumentation is not very thorough. It would be better to refer to the vertical shear of the horizontal wind. We replaced this passus of the manuscript referring to Jacobi et al. (2015) by a text referring to Jaen et al. (2023), who showed that the wind shear has its maximum during summer.

L 576

'we observe in our data changes from south in spring to east in winter'. I cannot read the direction is south in spring from Figure 15. Please check it.

Thanks for bringing our attention to this part. One cannot see this from in figure 15 (which is the natural choice when reading the manuscript) but must refer to figure 12. We added this info and tried to make the info a little bit more precise.

Figure 15.

Please indicate the location of center of the plot, as well as WE and NE line. Is 'zero' value shifted from the center, which is my guess from the scale axis? If so, what is the reason?

You are right, the "zero" value is shifted from the centre and there is no specific reason for it. We made some modifications and hope that the figure is now clearer for you.